# More frequent extreme climate events stabilize reindeer population dynamics

Brage B. Hansen [1], Marlène Gamelon [1], Steve D. Albon [2], Aline M. Lee [1], Audun Stien [3], R. Justin Irvine [2], Bernt-Erik Sæther [1], Leif E. Loe [4], Erik Ropstad[5], Vebjørn Veiberg [6] & Vidar Grøtan[1]

Extreme climate events often cause population crashes but are difficult to account for in population-dynamic studies. Especially in long-lived animals, density dependence and demography may induce lagged impacts of perturbations on population growth. In Arctic ungulates, extreme rain-on-snow and ice-locked pastures have led to severe population crashes, indicating that increasingly frequent rain-on-snow events could destabilize populations. Here, using empirically parameterized, stochastic population models for High-Arctic wild reindeer, we show that more frequent rain-on-snow events actually reduce extinction risk and stabilize population dynamics due to interactions with age structure and density dependence. Extreme rain-on-snow events mainly suppress vital rates of vulnerable ages at high population densities, resulting in a crash and a new population state with resilient ages and reduced population sensitivity to subsequent icy winters. Thus, observed responses to single extreme events are poor predictors of population dynamics and persistence because internal density-dependent feedbacks act as a buffer against more frequent events.

[1] Centre for Biodiversity Dynamics, Department of Biology, Norwegian University of Science and Technology, NO-7491 Trondheim, Norway. [2] James Hutton Institute, Aberdeen AB15 8QH, UK. [3] Norwegian Institute for Nature Research, Fram Centre, NO-9296 Tromsø, Norway. [4] Norwegian University of Life Sciences, NO-1432 Ås, Norway. [5] Norwegian University of Life Sciences, NO-0033 Oslo, Norway. [6] Norwegian Institute for Nature Research, NO-7485 Trondheim, Norway. These authors contributed equally: Brage B. Hansen, Marlène Gamelon. Correspondence and requests for materials should be addressed to B.B.H. (email: brage.b.hansen@ntnu.no) or to M.G. (email: marlene.gamelon@ntnu.no)

Extreme climate events can induce severe population crashes and destabilized population dynamics across animal taxa and biomes[1–4]. For instance, El Niño years severely affect the dynamics of birds[5], and droughts have led to extinctions in butterflies[6]. As accumulated evidence now suggests that global warming comes with an increase in the frequency of extreme climate events[7–9], their ecological impacts are also receiving more attention[4], yet mostly in plants. Given the inherent rareness of extreme events and the anecdotal approaches to study them in animals, the scientific focus has almost exclusively been on single events and their short-term effects[10]. However, especially in long-lived species where intrinsic properties regulate population dynamics, the impact of an environmental perturbation could vary with how often it occurs[11]. Thus, ignoring longer-term impacts, as well as anticipated changes in event frequencies under global warming[9], may ultimately lead to biased predictions of population persistence[4].

As some types of (previously extreme) events become progressively less rare—typically in climate change hotspots, such as the Arctic[12]—novel opportunities for mechanistic insights and a predictive understanding of their ecological impacts are now emerging. In the Arctic, extreme warm spells and rain-on-snow (ROS) events in winter may cause impenetrable snow-packs and even encapsulate the entire vegetation in thick ground-ice[13]. Such environmental perturbations on the tundra are no longer that rare[12–15], and population crashes and destabilized dynamics linked to icing events have been reported for a range of herbivore species[2], including muskoxen *Ovibos moschatus*[14,16], caribou, and wild reindeer *Rangifer tarandus*[17–19]. One intuitive, yet perhaps naive, extrapolation from observations of single population crashes following extreme ROS events would be to expect more variable dynamics and greater extinction risk with continued warming. However, theoretical studies indicate that this may not necessarily be the case[11]. The effects of environmental stochasticity on population growth can be buffered by density-dependent feedbacks[20–23]. Perturbations may have little or no impact at low density when resource competition is weak[24]. In addition, population responses to environmental stochasticity can depend on demographic structure, with some age classes being less sensitive to environmental fluctuations than others[20,25]. A change in population structure towards more resilient age classes after a population crash, i.e. a new population state[26], may therefore promote positive population growth rates and reduce the probability of new crashes in subsequent years. Thus, it has been suggested that high frequencies of bad years may lead to less variable population growth and, hence, stabilized rather than destabilized population dynamics[11]. However, empirical support for this prediction is still lacking.

Here, based on demographic population modelling of empirical time-series data[27], we evaluate how changes in the frequency of rainy and icy winters affect wild reindeer *R. t. platyrhynchus* population dynamics in Svalbard, a climate change hotspot in the High Arctic[13]. Because of the rapidly warming winter climate and the strong ROS signals in both reindeer demographic performance[28] and abundance[29], this northernmost ungulate represents an excellent case study for exploring the effects of more frequent extreme events. We show that the impact of an extreme ROS and icing event on reindeer survival, fecundity, and population growth rate is strongly age- and density-dependent. A population crash causes relaxation of density dependence, more resilient age structure, and, thereby, a long-lasting reduction in the population sensitivity to subsequent extreme events. Thus, because effects of environmental stochasticity are modified by internal density-dependent feedback, frequent extreme events dampen the population dynamics and even reduce the extinction risk.

## Results and discussion

**Exploring the climate–density interaction.** As a preliminary analysis, we first explored the impact of ROS and population density on annual reindeer population growth rates over the study period 1994–2014 (Fig. 1), obtained from the posterior means of an integrated population model (IPM) combining mark-recapture and count data[27,30] (Fig. 2). Because the trend for Arctic greening[31] due to gradually warmer and longer summers[32] is likely to influence the carrying capacity of the reindeer population[28], we accounted for variation in winter length and a linear change in carrying capacity. As expected, we found a strong negative effect of ROS on annual population growth rate (Fig. 1c

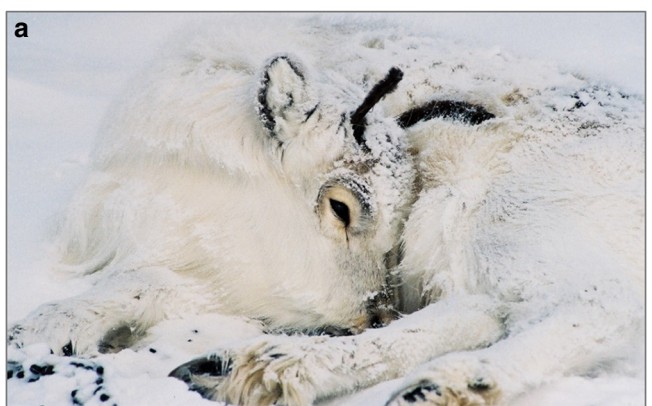

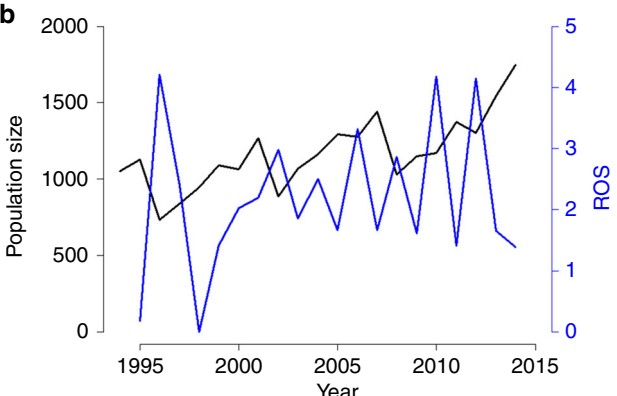

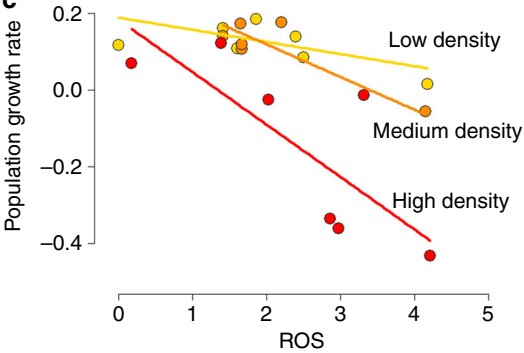

**Fig. 1** Density-dependent effects of rain-on-snow (ROS) and icing shape reindeer population fluctuations. **a** Occasionally, icing causes die-offs. This calf died from starvation in the 2001/2002 winter, which was characterized by high reindeer density and high ROS amounts. **b** Annual fluctuations in ROS (blue line) and total female population size *N* (black line) in the study period. **c** Density-dependent effect of ROS on the population growth rates (see Supplementary Table 1). For illustration, population densities (*N*) are classified into low (yellow symbols), medium (orange), and high (red)

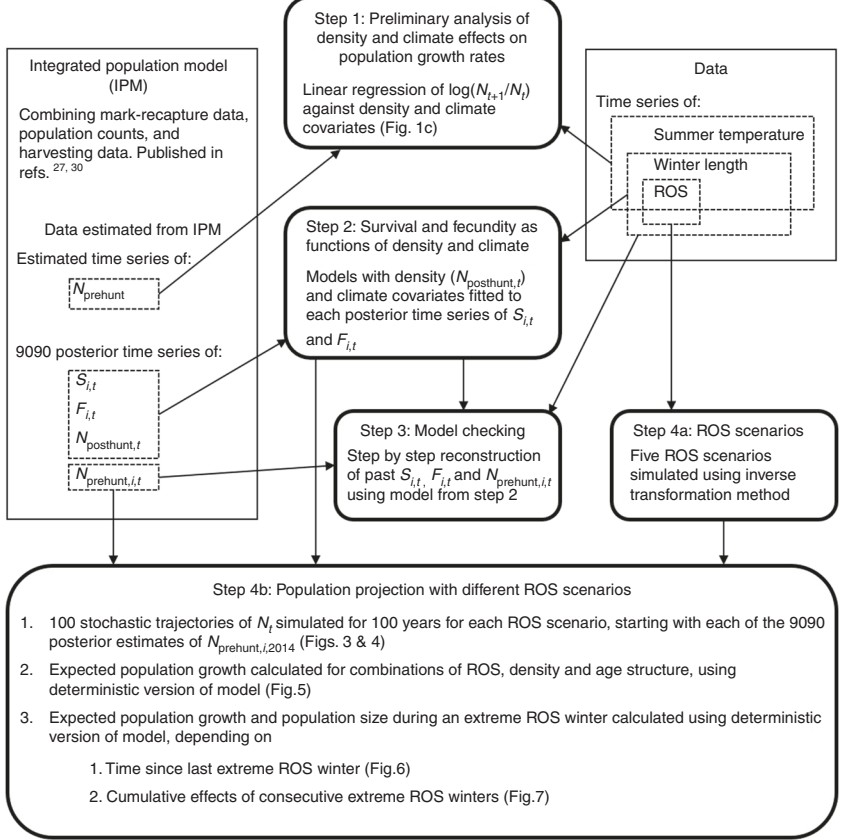

**Fig. 2** Schematic summary of the different analytical steps. Boxes with rounded corners show the analytical steps used in this paper. Square boxes contain data or estimates that are fed into the models in steps 1–4. $S_{i,t}$ and $F_{i,t}$ refer to survival and fecundity estimates for age class $i$ at time $t$, $N_t$ is the population size at time $t$, either age-structured before the hunting season ($N_{prehunt,i,t}$) or right after the end of the hunting season ($N_{posthunt,t}$), and ROS is a measure of rain-on-snow, as defined in the Methods. Note that in this figure, we use the term density in place of population size when considering density effects of population size on population growth (i.e., density dependence). This is to aid the reader in following how $N_t$ was used in different steps of our analysis (as a cause of density dependence, or as a response variable), and does not indicate a change in the way $N_t$ was estimated

and Supplementary Table 1). This effect was diminished at low densities, when food competition is weak or negligible (cf. ref. [33]) even under icy conditions.

**Accounting for age-specific effects of climate and density.** Second, based on these preliminary findings, we explored the underlying demographic mechanisms by modelling annual age-specific survival and fecundity rates, obtained for 9090 posterior samples from the IPM[27], as a function of weather and population size, allowing the effect of ROS to depend not only on density but also on age class[20] (Supplementary Figs. 1 and 2, Supplementary Table 2). As expected in long-lived ungulates[22,34], vital rates of young and old age classes were the most variable and the most strongly influenced by ROS events (Supplementary Table 2). Third, using these functions for survival and fecundity rates, together with past age structures (Supplementary Fig. 3) and weather conditions, we reconstructed fluctuations in vital rates and population sizes. Predicted population growth rates (Supplementary Fig. 4) were strongly correlated with observed growth rates (Pearson's correlation $r = 0.89$), suggesting high predictive power of the population model (Supplementary Fig. 5).

**Population dynamics and persistence under climate change.** We evaluated the demographic consequences of a set of ROS scenarios for the reindeer population, using stochastic simulations of the population model (Fig. 3). Global circulation models suggest that mid-winter warm spells and heavy ROS events will

become more frequent in the future Arctic[12,14,15], including Svalbard. Therefore, we varied the frequency of extreme ROS winters (see Supplementary Fig. 6) from very low (virtually never) to low, medium (as observed in the past, i.e. 1962–2014), high, and very high (the likely future scenario). A very high frequency of extreme ROS winters (rightmost panels in Fig. 3) reduces the mean population size by only 11% (Fig. 4a, Table 1) compared to the scenario describing observed historical conditions (i.e. medium ROS winter frequency, mid panels in Fig. 3) and by 25% (Fig. 4a, Table 1) compared to the very low frequency scenario (leftmost panels in Fig. 3). However, very frequent ROS winters also result in a strong reduction in the temporal variability in population sizes and growth rates (Figs. 3 and 4, Table 1). Such a change in population variability occurs because the impact of a bad year interacts with density and age structure (Fig. 5), which, in turn, are likely to be influenced by the time since the previous bad year. This also has implications for the frequency and magnitude of population crashes under different ROS scenarios (Fig. 3). Accordingly, the probability of going extinct (population size $N = 0$) during a period of 100 years is about 15,000 times higher for the medium ROS scenario, i.e. the observed historical climate, than for the very high ROS scenario anticipated under continued global warming (Table 1). Likewise, the probability of going quasi-extinct (here arbitrarily defined as $N < 100$) is 10-fold.

**Density-dependent feedback modifies climate change effects.** To illustrate how climate–demography interactions can lead to

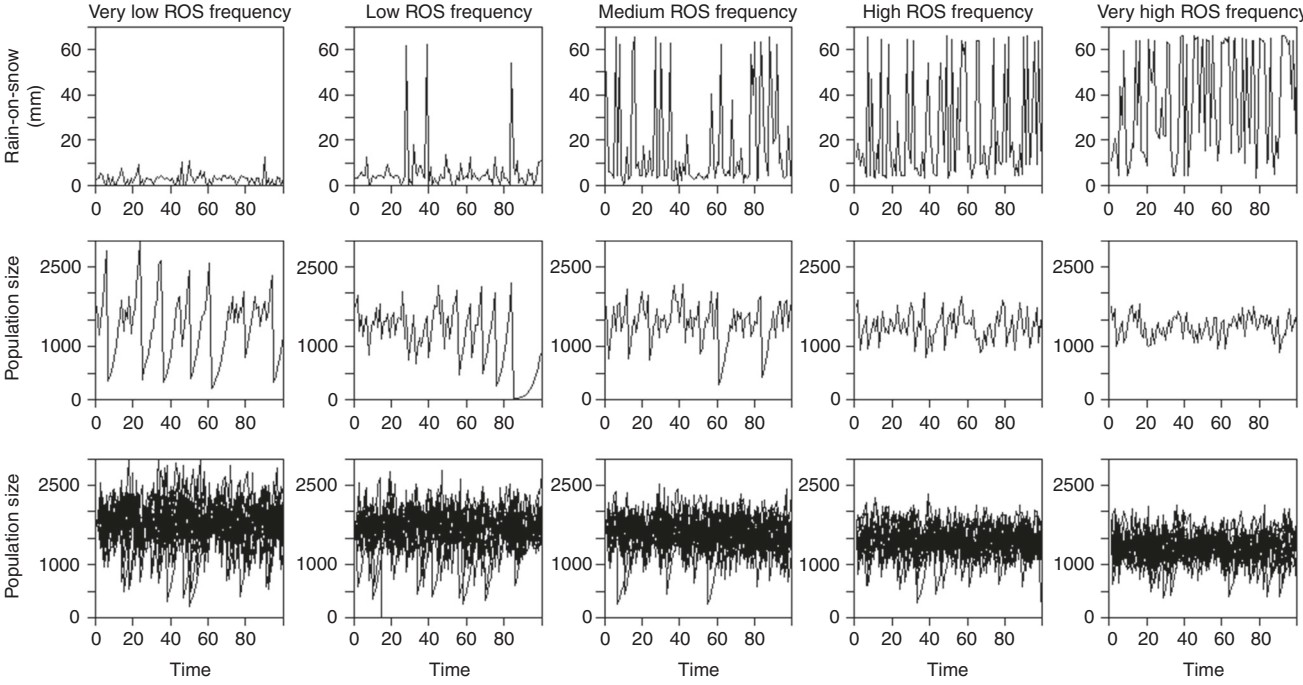

**Fig. 3** Reindeer population fluctuations under five different rain-on-snow (ROS) scenarios. Scenarios correspond to different distributions of observed values (1962–2014), with increased frequency of extreme ROS winters. Upper panels: One randomly selected ROS simulation for each of the five scenarios, which (from the left to the right) span from a very low to very high frequency of extreme ROS winters (see Supplementary Fig. 6). Mid panels: The stochastic simulation of the population dynamics for each scenario shown in the upper panels, based on age-structured density-dependent models of vital rates. Lower panels: Stochastic simulations of the population dynamics for ten randomly chosen simulations per scenario

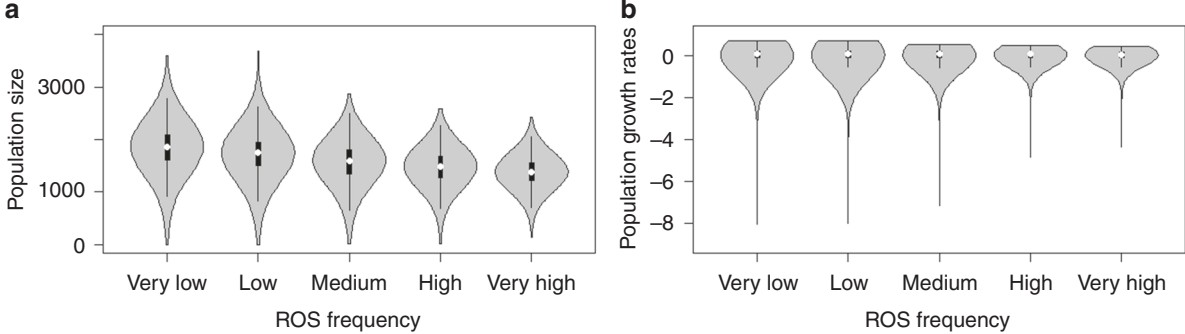

**Fig. 4** Reindeer population-dynamic parameters under five different rain-on-snow (ROS) scenarios. Scenarios correspond to different distributions of observed values (1962–2014), with increased frequency of extreme ROS winters. **a**, **b** Distributions of 10,000 random samples of **a** population sizes and **b** growth rates from stochastic simulations of the population dynamics for each scenario, based on age-structured density-dependent models of vital rates. White dots show the median, boxes show the lower and upper quartiles (25th and 75th percentiles), and whiskers extend to the most extreme data point which is no more than 1.5 times the interquartile range from the box

### Table 1 Reindeer population parameters under five different rain-on-snow (ROS) scenarios

| Population parameter | Frequency of extreme ROS winters | | | | |
| --- | --- | --- | --- | --- | --- |
| | **Very low** | **Low** | **Medium** | **High** | **Very high** |
| Mean, $N$ | 1820 | 1698 | 1546 | 1458 | 1369 |
| Variance, $N$ | 189,040 | 165,458 | 152,002 | 102,965 | 71,090 |
| Variance, growth rate | 0.100 | 0.099 | 0.106 | 0.079 | 0.062 |
| Quasi-extinction (%) | 4.65% | 4.59% | 4.10% | 1.25% | 0.41% |
| True extinction (%) | 0.84% | 0.45% | 0.30% | 0.02% | 0.00002% |

Scenarios correspond to different distributions of observed ROS values (1962–2014), with increased frequency of extreme ROS winters. True extinction risks and quasi-extinction risks are reported as the proportion of the 909,000 simulations (9090 population models [i.e. posterior samples] × 100 simulations) of population trajectories reaching population size $N = 0$ (during 100 years) or $N < 100$ (at least once during 100 years), respectively

such differences in long-term population dynamics and persistence under contrasting ROS scenarios, we ran deterministic population simulations where we increased the time elapsed between two extreme ROS winters (Fig. 6). We modelled the extreme ROS winter as the maximum ROS amount ever recorded at the local weather station (i.e. 1996, Fig. 1b). Provided a rather high initial density characterized by a low proportion of prime-aged animals (i.e. 3–8-year olds) in year $t = 0$ (i.e. representative of the population state before a known population crash in 1996), an extreme ROS winter leads to a dramatic decline in population size in year $t = 1$ (Fig. 6a). In particular, many calves and old animals die (Supplementary Fig. 1) and very few new calves are born (Supplementary Fig. 2) in crash years[27,28], also resulting in a marked increase in the proportion of prime-aged animals (Supplementary Fig. 3). With no subsequent ROS winter (Fig. 6a, leftmost panel), the low density (i.e. relaxation of density

dependence) and new age structure allow the population to recover towards an asymptotic population size (Fig. 6a) and stable age structure (Supplementary Fig. 7, leftmost panel). If a second ROS winter occurs immediately after the first one (Fig. 6a, second leftmost panel), the model predicts no additional decrease in population size before recovery commences. This occurs because the previous year's crash generated a new population state, with low density as well as large proportion of prime-aged animals, showing little sensitivity to ROS. This promotes population recovery (Fig. 5). If the second ROS winter is delayed until $t = 2$, the previous year's recovery in population size induces a slight population decline due to the harsh feeding conditions. Because of these lagged effects of a population crash, it takes about 7 years before the impact of an icy winter returns to the initial level at $t = 0$ (Fig. 6b). Accordingly, if there are several extreme ROS winters in a row, population size converges towards a reduced density of 1000–1500 individuals due to relaxation of density dependence (Fig. 7). These interactions between climate effects, density, and age structure explain why our study population did

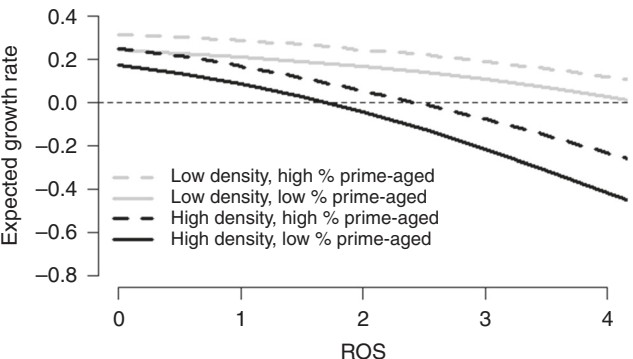

**Fig. 5** Expected population growth rate as function of rain-on-snow (ROS), density, and age structure. The relationship is shown for combinations of two contrasting initial population densities ($N = 1200$ and 1700) and two initial age structures (proportion of prime-aged [3–8-year olds] = 28 and 51%), illustrating why extreme climate events do not always lead to a large population decline. Horizontal dashed line denotes population growth rate = 0

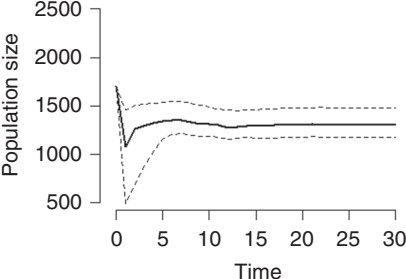

**Fig. 7** The effect of many consecutive extreme ROS winters on reindeer population size. Time-series of population size ($N$) are from deterministic simulations with rather high initial density $N$ (1700) and low initial proportion of prime-aged (28%), as in Fig. 6. Solid line shows the mean and dashed lines the 2.5th and 97.5th percentiles from 9090 population models (i.e. based on 9090 posterior samples)

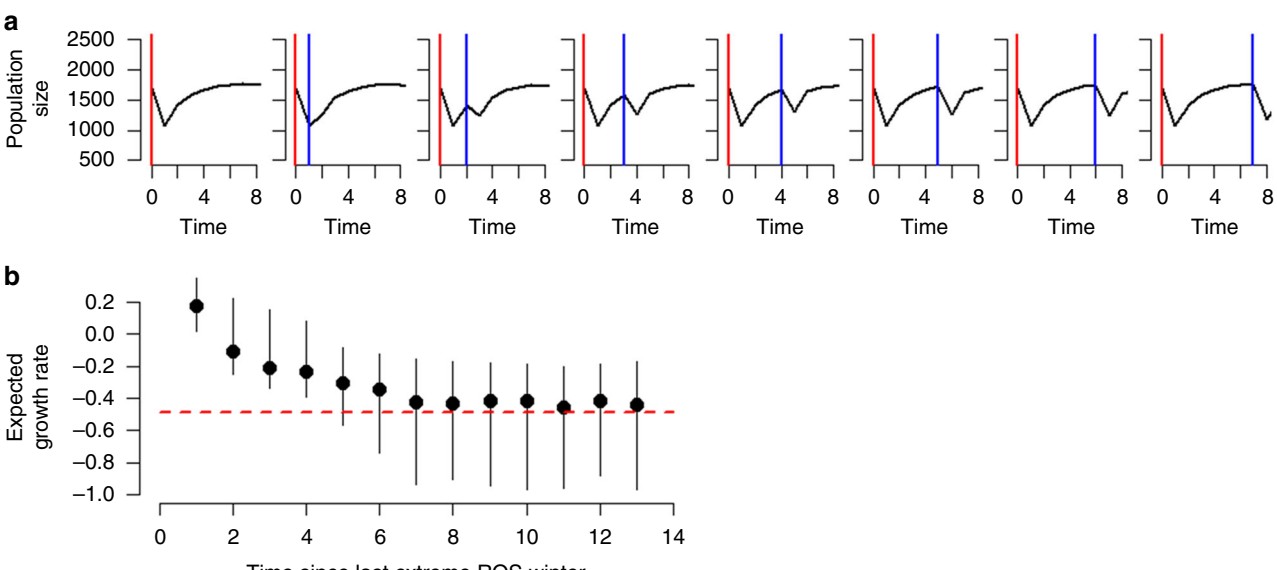

**Fig. 6** The time elapsed between two extreme rain-on-snow (ROS) winters influences reindeer population stability. **a** Population size ($N$) time-series from deterministic simulations with rather high initial density $N$ (1700) and low initial proportion of prime-aged (28%). The time elapsed from the first extreme ROS winter (red line) to the second one (blue line) increases from left to right. **b** Expected population growth rate for the second extreme ROS winter, plotted against the time elapsed since the first one. Circles show the mean and error bars the 2.5th and 97.5th percentiles from 9090 population models (i.e. based on 9090 posterior samples). Red dashed line shows the expected growth rate in the first extreme ROS winter

not crash in recent icy winters (Fig. 1b). Such lagged responses to environmental stochasticity are predicted by theory[11,22], yet poorly documented (but see[20]). Thus, to our knowledge, this study provides the first evidence from the wild that fluctuations in population size of long-lived species are dampened when extreme climate events become more frequent[11].

In contrast to our study area (a hotspot for winter climate change[13,14]), heavy ROS is still an uncommon phenomenon in many parts of the Arctic[14], where the anticipated near-future winter warming[12,15] may first imply moving from very low to low frequency of extreme ROS winters (Fig. 3). When ROS is still that rare, ungulate populations are more likely to be in the kind of state (in terms of density, as well as demographic structure) that may lead to a crash when an extreme event occurs. Recent observations of occasional population crashes across the circumpolar Arctic[14,17,18] support this, with potentially large socioeconomic and ecosystem implications[35]. However, one important message from our simulations, supported by observed population dynamics (Fig. 1b), is that the impact of an extreme ROS winter on the population growth rate is far less dramatic if it occurs soon after a previous perturbation (i.e. moving towards high and very high ROS frequencies, Fig. 3). This is a likely characteristic of the near-future winter climate in many coastal Arctic regions[14,15], including Svalbard.

Extreme environmental perturbations can have unexpected ecological impacts[36], and therefore represent one of the major challenges in future predictions of ecosystem change[1,37]. Because extreme climate events are by definition rare, studies based on empirical data from the wild are also typically of anecdotal nature[10]. Our case study clearly demonstrates that population-dynamic inference based solely on single events and their short-term impacts—ignoring potential long-term impacts, as well as the consequences of multiple events—may lead to erroneous conclusions. In particular, our results emphasize how internal density-dependent feedback processes can modify the effects of environmental stochasticity and, hence, buffer populations of long-lived species if extreme events become the norm due to global warming.

## Methods

**Study area and species**. Our study population of wild Svalbard reindeer is located in the Reindalen, Semmeldalen, and Colesdalen valley system in central Spitsbergen (78°N, 15°E), Svalbard, Norway. The area is characterized by U-shaped coastal valleys with High Arctic tundra vegetation of low stature, dominated by mosses, graminoids, dwarf shrubs, and forbs. Because of semi-isolation by the sea, steep mountains, and glaciers, as well as a stationary behaviour, there is very little exchange of animals with other nearby reindeer populations[38]. The reindeer occur alone or in small groups and are not subject to significant levels of predation (polar bear attacks are very rare[39]), inter-specific competition, or insect harassment. Each fall, there is a very low level of harvest[40], and some reindeer have been culled for scientific purpose[41]. Both hunted and culled animals are reported and their age is determined[42].

ROS has previously been reported as the main climatic driver of vital rates and population growth in Svalbard reindeer[2,19,29,40]. The proximate mechanism behind this is that ROS causes icing, which restricts food availability, in turn affecting body mass loss during winter[28]. During the study period, summers have become warmer causing increased plant productivity[2,32,43,44], representing the most plausible explanation for positive trends in autumn body mass and population size (Fig. 1b) in our study population[28].

**Data**. The reindeer data used in this study originate from a mark-recapture study, which protocol applies with and is approved by the Norwegian Animal Research Authorities and the Governor of Svalbard. The reindeer are captured as calves (and recaptured in later years) in a net between two snow mobiles during April each year, and a post-breeding resighting survey is performed in early August[40]. A posterior sample of 9090 estimates of annual survival (Supplementary Fig. 1) and fecundity (Supplementary Fig. 2), as well as population size ($N$), for six female age classes during 1994–2014, were obtained from an IPM[27,30,45] (Fig. 2). The posterior means of each annual demographic parameter are hereafter referred to as observed values or observations. The six age classes considered are 0 (calves), 1, 2, 3–8, 9–11, and ≥12 year olds, corresponding to age classes 1–6 in Supplementary Figs. 1 and 2,

Supplementary Table 2. The model combines individual mark-recapture data with population counts within a Bayesian state-space modelling framework, accounting for observation error and demographic stochasticity. The small and controlled level of harvest and scientific culling was also accounted for. Detailed modelling and data methodology are described in ref. [27], with further model updates in ref. [30].

We obtained daily historical weather data for Longyearbyen airport, ca 25 km from our study population, from the Norwegian Meteorological Institute (freely available at http://eklima.met.no). Based on daily mean temperatures and precipitation during 1962–2014, we first calculated annual winter rain amount by summing the amount of precipitation recorded at temperatures ≥1 °C during November–April[2,40]. We added 1 to this value and log-transformed it to get the variable ROS, an icing/winter harshness proxy[29] (Fig. 1b). Second, we estimated the length of the winter based on a 10-day running mean of daily temperatures. We considered the onset of winter as the first day in autumn when the running mean was <0 °C and then stayed <0 °C for a minimum of 10 consecutive days. Likewise, end of winter was estimated as the first day (after the winter period November–April) when the running mean was >0 °C and then stayed >0 °C for a minimum of 10 consecutive days.

**Population growth rate as a function of density and climate**. Based on the observed total population sizes ($N_t$, where $t$ is time) from the IPM[27,30], we first explored climate–density interactions by fitting a linear regression of annual population growth rate ($\log(N_{t+1}/N_t)$) against $\log(N_t$, linearly detrended) and climate covariates, namely ROS and length of the winter (Fig. 2). We found a negative effect of length of the winter ($t$ to $t+1$) and a negative interaction effect between the amount of ROS ($t$ to $t+1$) and population size (Supplementary Table 1), suggesting a density-dependent ROS effect and supporting the assumption that increased plant abundance due to warmer summers[32,44] has a gradual effect on the carrying capacity $K$ (thereby, the detrended $N$). To visually illustrate the ROS–density interaction effect (Fig. 1c), we divided population sizes into high, medium, and low $N$ (detrended). The respective linear regression model replacing the covariate $N$ with these population size classes (as factor) provided qualitatively similar results as the model described above (Supplementary Table 1).

**Survival and fecundity as functions of density and climate**. We obtained 9090 posterior samples from the IPM[27,30], each of them consisting of annual age-class-specific demographic rates (i.e. survival and fecundity) and population sizes from 1994 to 2014. For each of these posterior samples, we investigated the effects of weather and population density on survival and fecundity, generating in total 9090 population models. First, the effects of weather and population density on survival $S$ of each age class $i$ at year $t$ were estimated separately for each posterior sample (Fig. 2, Supplementary Fig. 1), using linear mixed-effects models (function $lme$ in R package $nlme$). As survival might be negatively affected by an increase in population size $N_{posthunt}$ (i.e. number of calves + adult females at the end of the hunting season), we tested for an effect of (scaled) population size in year $t$ on survival. We fitted a density-dependent ROS effect[20] (cf. Fig. 1c) using the form $ROS'_t = ROS_t \times e^{k \times N_{posthunt,t}}$ (see ref. [21] for a similar approach). This form of the interaction effect ensures that the effect of ROS is strictly negative or positive (depending on the fitted coefficient of ROS') for all values of $N_{posthunt}$ (Supplementary Figs. 8 and 9). In contrast, the simpler and more common specification of an interaction effect, such as $k \times ROS_t \times N_{posthunt,t}$, would implicitly lead to a switch in sign of the effect of ROS at some level of $N$, and the stronger the interaction effect, the more likely it is that the change of sign is well within data range. Note that $k$ was estimated using an optimization function aiming at minimizing Akaike's Information Criteria. Moreover, mean survival is likely to differ among age classes, and the effect of climate may also differ among age classes (e.g. ref. [20]), thus we included an interaction between age and the density-dependent ROS effect. Finally, year was added as a fixed (numeric) effect to account for trends. Year was also added as a random (intercept) effect to account for non-independence among age classes, leading to a survival model with the following form:

$$\text{logit}(S_{i,t}) = a_i + N_{posthunt,t} + \text{year} + ROS'_t + ROS'_t \times a_i \\ + \text{length\_winter}_t + \text{random(year)} + \varepsilon_{S_{i,t}} \quad (1)$$

where $a_i$ is the age class $i$, $N_{posthunt,t}$ is population size just at the end of the hunting season, and $\varepsilon_S$ is the residuals of the regression. The posterior distributions of the parameters of Eq. (1) shown in Supplementary Table 2 represent a combination of uncertainty due to finite time series, uncertainty stemming from stochasticity, and uncertainty in the IPM. Likewise, we ran a model of similar structure for fecundity $F$ of each age class $i$ at year $t$, as females produce at most one calf per year (Supplementary Fig. 2). Note that there is one age class less than in the survival analysis, as calves do not get pregnant:

$$\text{logit}(F_{i,t}) = a_i + N_{posthunt,t} + \text{year} + ROS'_t + ROS'_t \times a_i \\ + \text{length\_winter}_t + \text{random(year)} + \varepsilon_{F_{i,t}} \quad (2)$$

Again, the posterior distributions of the parameters of Eq. 2 shown in Supplementary Table 2 represent a combination of uncertainties (see above for survival). Six and five age classes were considered for survival and fecundity, respectively[27,30]. We assumed equal survival and fecundity among ages 3–8 years, among ages 9–11 years, and among ages >11 years.

**Reconstructing past vital rates and population growth rates**. Based on the above models of vital rates, for each posterior sample we estimated age-specific survival and fecundity rates for past conditions of population density, winter harshness (i.e. ROS amount), and winter length (Fig. 2, Supplementary Table 2). Importantly, to account for sources of environmental stochasticity due to processes other than covariates included in the model, we estimated a covariance matrix $\Sigma$ of the different vital rates (fecundity, survival) for all age classes based on the random year effects and the residuals $\varepsilon_{S_{i,t}}$ and $\varepsilon_{F_{i,t}}$ of Eqs. 1 and 2. From this covariance matrix $\Sigma$, we generated 100 new residuals from a multivariate normal distribution.

These rates then allowed the population size at time $t+1$ to be estimated from the population size of each age at time $t$. The population size just before the hunting season at time $t+1$, $N_{\text{sim},t+1}$, corresponds to the sum of females of different ages:

$$N_{\text{sim},t+1} = N_{\text{sim calves},t+1} + N_{\text{sim yearlings},t+1} + N_{\text{sim 2},t+1} + \cdots + N_{\text{sim 13},t+1} \quad (3)$$

Each of these terms can be estimated based on the vital rates. The number of calves produced, $N_{\text{sim calves},t+1}$ (first term on right side of Eq. 3), consists of calves produced by females of each age (except yearlings). $N_{\text{sim calves},t+1}$ was modelled using a binomial process to allow for demographic stochasticity (i.e. chance events that affect individuals independently):

$$N_{\text{sim calves},t+1} \sim \text{Bin}\left(N_{\text{sim 2},t+1}, F_{\text{sim 2},t}\right) + \text{Bin}\left(N_{\text{sim 3},t+1}, F_{\text{sim 3},t}\right) \\ + \cdots + \text{Bin}\left(N_{\text{sim 13},t+1}, F_{\text{sim 13},t}\right) \quad (4)$$

where $F_{\text{sim }i,t}$ is the estimated fecundity rate for females of age $i$ at time $t$, resulting in calves at time $t+1$. For instance, $F_{\text{sim 2},t}$ is the probability of a 2-year-old female at time $t$ having a calf at heel at time $t+1$ (depending on climate and population size at time $t$, see Eq. 2). The number of female calves was then drawn from a binomial distribution with a probability 0.5 (i.e. we assumed a balanced sex ratio).

$N_{\text{sim yearlings},t+1}$ (second term on right side of Eq. 3) corresponds to the number of female calves that have survived from time $t$ to time $t+1$ and was also modelled using a binomial process to include demographic stochasticity. Moreover, a few female calves are removed from the population by hunting ($H$) or scientific culling ($C$). Thus the number of female yearlings is modelled as follows:

$$N_{\text{sim yearling},t+1} \sim \text{Bin}\left(N_{\text{sim calves},t} - H_{\text{calves},t} - C_{\text{calves},t}, S_{\text{sim 1},t}\right) \quad (5)$$

where $S_{\text{sim }i,t}$ is the estimated survival probability of females of age $i$ at time $t$. Similarly, $N_{\text{sim 3},t+1}$, …, and $N_{\text{sim 14},t+1}$ (all other terms in Eq. 3) correspond to the population size in the previous age that have survived from time $t$ to time $t+1$. Thus, for age $j \in [2, 13]$:

$$N_{\text{sim }j+1,t+1} \sim \text{Bin}\left(N_{\text{sim }j,t} - H_{j,t} - C_{j,t}, S_{\text{sim }j,t}\right) \quad (6)$$

Note that summer mortality in calves (as well as for other age classes) is considered to be close to zero[46]. Thus $N_{\text{sim},t+1}$ can be estimated from observed values of ROS, length of the winter, and population size, estimating $S$ and $F$ from the models presented above.

Using the models described above, we estimated age-specific survival rates (from 1994 to 2013), fecundity rates (from 1995 to 2014), and the population size (from 1995 to 2014) through a step-by-step approach. Each year was estimated based on observations of age-specific population sizes (provided by the IPM; Supplementary Fig. 3) and observed ROS and winter length the previous year. Thus, in Eqs. 5 and 6, $N_{\text{sim calves},t}$ was replaced with $N_{\text{obs calves},t}$ and $N_{\text{sim }j,t}$ was replaced with $N_{\text{obs }j,t}$. The estimated population size $N_{\text{sim},t+1}$ may then be calculated step by step (Eq. 3) and compared with the observed population size.

The annual age-specific survival rates (Supplementary Fig. 1) and fecundity rates (Supplementary Fig. 2) estimated within this framework were closely correlated to the ones observed (i.e. obtained from the IPM). Pearson's correlations ranged from $r = 0.72–0.89$ for survival rates and $r = 0.65–0.84$ for fecundity rates. Accordingly, our model was able to reconstruct annual fluctuations in total population size well (Supplementary Fig. 4), with a strong correlation ($r = 0.89$) between estimated and observed population growth rates (Supplementary Fig. 5).

**Reindeer population projections with different ROS scenarios**. Finally, we analysed how increasing frequency of rainy winters affects the population growth rate (Fig. 2). To simulate realizations of ROS (Supplementary Fig. 6), we used the observed ROS data from 1962 to 2014 and an inverse transformation method. Given a continuous uniform variable $U$ in [0,1] and a cumulative distribution function $D$, the random variable $X = D^{-1}(U)$ has distribution $D$. We can change the distribution (i.e. different scenarios of ROS) by using a non-uniform distribution of $U$. Thus, because the cumulative ROS distribution is bounded between 0 and 1, we used a beta distribution $f(U) \propto U^{\alpha_1 - 1}(1 - U)^{\alpha_2 - 1}$ where $\alpha_1$ and $\alpha_2$ are shape parameters to simulate five different ROS scenarios. We used the following parameters to obtain different ROS scenarios (Supplementary Fig. 6): very high frequency of extreme ROS winters ($\alpha_1 = 4$, $\alpha_2 = 1$), high frequency ($\alpha_1 = 2$, $\alpha_2 = 1$), low frequency ($\alpha_1 = 1$, $\alpha_2 = 2$), and very low frequency ($\alpha_1 = 1$, $\alpha_2 = 4$). In addition, the medium frequency scenario simulating the historical state (1962–2014) was obtained by setting $\alpha_1 = 1$ and $\alpha_2 = 1$ leading to a uniform distribution as a special case of the beta distribution. Note that $0 < U < 1$ for all scenarios. Accordingly, our simulations

(Fig. 3, Supplementary Fig. 6) are conservative as the simulated realizations will fall within the range of observed (historical) values of ROS.

Using the estimated mean age-specific population sizes in 2014 and each of our 9090 age-structured population models (i.e. based on 9090 posterior samples, see Modelling step 2), we simulated for each of the five ROS scenarios (i.e. distributions) 100 stochastic population size trajectories for 100-year projections. This resulted in 4,545,000 population size time-series (454,500,000 population sizes) in total. In other words, instead of using a step-by-step approach (as described above), we fed our age-structured population model with the observed age-specific numbers in 2014 and the ROS trajectories. Note that, for each simulated trajectory, winter length was drawn from a normal distribution with the mean and the standard deviation observed between 1994 and 2014, i.e. our reindeer study period.

We explored how increasing amount of ROS affects the population growth rate, under different age structures and population densities. To do so, we considered two initial age structures (low prime-aged [3–8-year olds] proportion = 28%, high prime-aged proportion = 51%). Note that the two age structures reflect the observed demography before and after the 1996 population crash (Fig. 1b, Supplementary Fig. 3), respectively. For each age structure, we considered two initial population densities (low $N = 1200$, high $N = 1700$). We simulated an increasing amount of ROS (from 0 to 4.2, on a log-scale) from the deterministic version of our population model, keeping winter length constant at its mean observed value (1994–2014). From these population projections, the expected population growth rate for each ROS value was determined for each combination of density/age structure (see Fig. 5).

Finally, we examined how the time elapsed since the previous extreme ROS winter affects the population growth rate during a second extreme ROS winter. An extreme ROS winter reflects the conditions in 1996 (i.e. 1995/96), with the highest ROS value recorded (log ROS = 4.2). We simulated different ROS sequences where ROS was kept at its mean recorded value (1994–2014) except for a first extreme ROS winter always occurring at $t = 0$, and a second extreme ROS winter occurring at $t = 1$, then at $t = 2$, and so on until $t = 7$. We also included a ROS sequence with no second extreme ROS winter and a sequence with many consecutive extreme ROS winters. Keeping winter length constant at its mean observed value, we then used the deterministic version of our population model to calculate time-series of population sizes (Figs. 6a and 7) and age structures (Supplementary Fig. 7) following a rather high initial population density ($N = 1700$) and a low initial proportion of prime-aged individuals (28%), reflecting the population state prior to the 1996 crash. We fed our population model with the eight different sequences of ROS and estimated the expected population growth rate for the second extreme ROS winter (Fig. 6b). All analyses were performed with the statistical software R[47].

**Reporting Summary**. Further information on experimental design is available in the Nature Research Reporting Summary linked to this article.

## Code availability
All computer code is available from the authors on reasonable request.

## Data availability
All data are available from the authors on reasonable request.

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

## Acknowledgements

This study was funded by the Research Council of Norway (RCN: POLARPROG project 216051, Centre of Excellence project 223257, KLIMAFORSK project 244647, and FRI-MEDBIO project 276080). Data collection was financed by the Norwegian Institute for Nature Research, Macaulay Development Trust, James Hutton Institute, UK Natural Environment Research Council (NERC), University Centre in Svalbard (UNIS), and RCN. We thank the Norwegian Meteorological Institute for weather data, the Governor of Svalbard for permission to do field work, and UNIS for logistics support.

## Author contributions

S.D.A., R.J.I., E.R., A.S., L.E.L. and V.V. collected reindeer data. B.B.H. initiated the study. M.G., V.G. and B.B.H. conducted the analyses. B.B.H. and M.G. wrote the first version of the manuscript and S.D.A., A.M.L., B.E.S., R.J.I., E.R., A.S., L.E.L., V.V. and V.G. contributed to discussions and revisions on later manuscript versions.

## Additional information

**Competing interests:** The authors declare no competing interests.

