## [Peer Review File · Nature Communications]

Reviewers' Comments:

Reviewer #1:

Remarks to the Author:

This study provides a fascinating exploration of feedback mechanisms between density dependence, environmental canalization or demographic buffering, and implications of climate change. I found the methods robust and the results extremely novel, interesting, and with major implications for understanding population dynamics in changing environments. I have only a few clarifications on the text and the methods, which I hope will help improve this very interesting manuscript.

The sentence in line 84-85 is a bit confusing... I would imagine that the posterior samples include the vital rates simulated from the IPM?

Lines 107 to 111: This sentence is a bit hard to grasp. Still, if I understand well, I find it interesting that the ratio is 10 for the quasi-extinct and 15,000 for the extinct. That suggests that many populations in the high frequency scenario still fall below 100 individuals, but that they avoid XXXXX

In lines 141-142 you state that when ROS is rare, populations are likely to be in a 'vulnerable' state. What do you mean by this? Is the 'vulnerable' state due to high densities and therefore high density dependence?

In equations (1) and (2) you include a fixed year effect and then a random(year) effect(?). This seems not only unnecessary but incorrect. I suspect that I am missing something here...

In lines 238-242 you use a logit link for fertility, so this implies that you assume that every female will produce at most one calf per year, is this realistic?

In lines 244-249 you explain that you pooled ages into general age-classes. How do you think this can affect your simulations? I ask because it is well known that reducing ages into age classes dampens down the variance in population growth rates...

In lines 254-258 you describe the covariance matrix for the vital rates, which you base on the random year effects and the residuals. How do you populate this covariance matrix, by calculating the covariation on the random terms between vital rates?

Lines 265-269: if I understand well and as you write it in eq (4), the variance of the demographic stochasticity is not inversely proportional to the population size, isn't it?

Line 278, when you describe the interval, replace the colon by a coma (i.e. [2, 13]).

In lines 287-289 if I interpret correctly these two sentences, you did not simulate the number of calves and 11+ year olds? Why?

In lines 300-302 you use F in reference to a distribution while you also use it when you refer to fecundity. For clarity, I would suggest that you change this.

In Lines 303-309 you say that you sampled from a Beta distribution. I'm curious, why a Beta distribution. Are the ROS values bound between 0 and 1? Also, if you simply randomly sampled from a Beta distribution the whole explanation of the inverse sampling in lines 300-302 is not necessary.

Lines 315-317: This passage is a bit confusing. Do you mean that you started all your simulated populations with the population numbers of 2014? Also, 100 simulated populations seems a bit low...

In Extended data figure 4, you include error bars to the observations. Does that mean that you are accounting for observation errors?

Reviewer #2:

Remarks to the Author:

This paper examines the population impact of an extreme environmental stressor (rain on snow, which causes icing conditions making subsequent feeding more difficult) on the dynamics of a reindeer population. Despite substantial increases in overwinter mortality associated with ROS, the authors use models parameterized for this population to project that increasing frequency of ROS (a likely outcome of climate change) will lead to a *reduction* in population fluctuations and extinction risk. This appears to be the result of a combination of demographic and density-dependent buffering, which has been examined in theoretical models, but not (to my knowledge) in particular populations.

The authors performed a thorough and well-thought-through analysis, and I detected no fundamental flaws. However, there are a number of areas where the methods and results are not presented in sufficient detail to fully rule out potential challenges to the conclusions. I detail these below. If the analysis stands up to this increased scrutiny, then I think it will be a valuable contribution to the literature on ecological responses to a changing climate.

The key relationships that lead to the result described above were not observed directly in the population. Indeed, a key factor in the mechanism--density dependence in the absence of ROS--is not supported by the population data (the curves for high, medium and low density intersect at $ROS=0$ in fig 1c; the main effect of N on population growth rate is indistinguishable from zero (Table ED1). Thus, the analysis depends critically on a previously parameterized model of the population, which is described elsewhere.

As a consequence, the ability to use the analysis to draw conclusions about the Svalbard reindeer herd (as opposed to more theoretical conclusions based on a "realistic" population model) requires that the model is an accurate and precise representation of the population processes that are important in the current context. The publications describing the model have been peer reviewed, and I have no reason to question the model (although my experience is that journal peer review provides no guarantee that a published ecological model is reliable); but any model is an approximation to reality, and its utility can only be judged in the context of a particular question. Thus, my question for the authors is how they have evaluated the model's suitability for the present purpose. The analysis summarized by ED Fig 4 provides some evidence that the model tracks the year-to-year variation in population abundance (although the confidence intervals for the observed abundances are generally quite large, resulting in low power, and in some years the model prediction deviates substantially from the point estimate of abundance). However, it would be helpful to have a qualitative assessment of whether any structural decisions made during the original model development might be more consequential in the current context than the original one, and explicit treatment of the effects of parameter uncertainty (it might be that the posterior samples in the second step are doing this, but it is unclear; see below).

From a cursory examination of the cited papers, it appears that ROS is not in the original IPM. Please explain how ROS data was matched with the IPM outputs to allow the regressions.

What constitutes a posterior sample from the IPM? A single year of vital rate estimates and associated covariates? A full time series of such values?

How were models in equations 1 and 2 estimated? If ROS came from observed data, then there really are only 20 independent observations, and there is a risk of pseudoreplication if the posterior samples are treated as independent. I would imagine that the ideal approach would be for each posterior sample to be a full time series, and the regression equations were estimated separately for each sample. The posterior distributions on the parameters of eqs. (1) and (2) would then represent a combination of uncertainty due to finite time series and uncertainty stemming from both stochasticity and parameter uncertainty in the IPM. Is this what was done? If so, please describe it explicitly. If not, please describe and justify the approach.

I don't understand the argument justifying the form of ROS' (ll. 225-7). The direction of the effect depends on the sign of the regression coefficient, which is unconstrained. Given that this introduces a very specific form of nonlinearity that contrasts with linear direct effect of N, this needs conceptual or empirical justification. Furthermore, the posterior CI for k spans zero, indicating that the interaction between N and ROS could be either positive or negative.

It doesn't appear that the simulations leading to figs 2 and 3, and to the primary result of the paper, account for uncertainty in the parameter estimates shown in Table ED2. Especially given that some of the estimates (notably the interaction between ROS and N) have credible intervals that span zero, I think that it would be critical to evaluate how robust the results are to that uncertainty.

Is the residual variance (attributed to other sources of environmental stochasticity) kept constant with increasing ROS frequency? Or is it rescaled to keep total variability constant? If the latter, that could contribute to result of lower population variability/extinction risk at high ROS frequency.

The variable "ROS" is defined and used inconsistently. In figure 1, ROS is shown as a quantity measured in mm. However, in the data description (ll. 195-6), ROS is defined as the log of (amount in mm +1). It is unclear which is being used to define ROS' at l. 226. Furthermore, in the simulation analysis, the authors refer to "ROS frequency scenarios" (e.g. ll. 103-111). As best I can tell from fig ED 6, these scenarios have different mean ROS intensity, along with shifts in the tails of the ROS intensity distribution. These inconsistencies and impressions in notation make the work difficult to follow.

The parameter values in table ED 2 are very difficult to interpret, largely because the "intercept" represents the demography of the first age class extrapolated back to the year 0 (nearly 2000 years before the beginning of the time series). Because of the positive coefficient for year, this results in rates on the probability scale of essentially zero, and greatly inflates the apparent uncertainty in first age class demography. I recommend either centering the year variable or rescaling it so that "year 0" is the beginning of the time series. It would also be helpful to have a version of the table that reported the age-class estimates directly (both the base survival/fecundity and the impacts of ROS'), rather than as differences from the first age class. This would reveal, for example, that the estimated effect of ROS' on age class 3 survival has a CI that spans zero.

The deterministic model results provide valuable insights, thank you for including this analysis. However, it seems to me that the result is not just (or even primarily) the fact that prime age individuals are less sensitive to ROS (which is only true for survival, incidentally; the fecundity ROS coefficients are more negative for prime age classes), but also due to the relaxation of density dependence. This should be discussed more thoroughly. Furthermore, since ROS disproportionately impacts age 1 survival, repeated high ROS years will substantially reduce recruitment to the prime

age population. If ROS events are simulated not just twice in a row, but many times in a row, does the population converge to a low-density equilibrium, or does it continue to decline? Also, are the deterministic model results robust to the uncertainty in the parameter estimates?

Revision of the manuscript NCOMMS-18-26911 “When the exception becomes the norm: more frequent extreme climate events stabilize ungulate population dynamics”, by Hansen, Gamelon et al.

We thank the Editor as well as the two reviewers for their careful review of our manuscript. Our paper was substantially amended, and we provide below a detailed description of changes made and a point-by-point response to comments and suggestions made by the reviewers (in italics font).

Reviewer #1 (Remarks to the Author):

This study provides a fascinating exploration of feedback mechanisms between density dependence, environmental canalization or demographic buffering, and implications of climate change. I found the methods robust and the results extremely novel, interesting, and with major implications for understanding population dynamics in changing environments. I have only a few clarifications on the text and the methods, which I hope will help improve this very interesting manuscript.

>> *We thank the reviewer for her/his positive evaluation of our paper.*

The sentence in line 84-85 is a bit confusing... I would imagine that the posterior samples include the vital rates simulated from the IPM?

>> *We agree with the reviewer that this sentence was confusing. It has been re-worded in the revised manuscript (see l. 82 p. 5 and l. 223 p. 11): “Second, based on these preliminary findings, we explored the underlying demographic mechanisms by modelling annual age-specific survival and fecundity rates, obtained for 9,090 posterior samples from the IPM²⁷, as a function of (...)” and “We obtained 9,090 posterior samples from the IPM^{27,30}, each of them consisting of annual age-class specific demographic rates (i.e. survival and fecundity) and population sizes from 1994 to 2014. For each of these posterior samples, we investigated the effects of weather and population density on survival and fecundity, generating in total 9,090 ‘population models’.”*

Moreover, we have added a new figure (Fig. 2), summarizing the different methodological steps. This schematic clarifies that we have estimated the effects of winter length, the amount of rain-on-snow, and population density, on the annual age-specific survival and fecundity rates previously obtained for the same population and study period (Lee et al. 2015 and Bjørkvoll et al. 2016).

Lines 107 to 111: This sentence is a bit hard to grasp. Still, if I understand well, I find it interesting that the ratio is 10 for the quasi-extinct and 15,000 for the extinct. That suggests that many populations in the high frequency scenario still fall below 100 individuals, but that they avoid XXXXX

>> *We clarified this by splitting up the sentence (l. 188 p. 6): “Accordingly, the probability of going extinct (population size $N = 0$) during a period of 100 years is about 15,000 times higher for the medium ROS scenario, i.e. the observed historical climate, than for the very high ROS*

scenario anticipated under continued global warming (Table 1). Likewise, the probability of going quasi-extinct (here arbitrarily defined as $N < 100$) is ten-fold.”

In lines 141-142 you state that when ROS is rare, populations are likely to be in a ‘vulnerable’ state. What do you mean by this? Is the ‘vulnerable’ state due to high densities and therefore high density dependence?

>> Thanks to the reviewer for pointing this out. We realise that the word “vulnerable” could be misleading. We reworded the sentence to (l. 144 p. 7): “When ROS is still that rare, ungulate populations are more likely to be in the kind of state (in terms of density, as well as demographic structure) that may lead to a crash when an extreme event occurs.”

In equations (1) and (2) you include a fixed year effect and then a random(year) effect(?). This seems not only unnecessary but incorrect. I suspect that I am missing something here...

>> Year was added as a fixed numeric effect to account for the temporal trend in survival and fecundity rates. Year has also been included as a random effect (factor) to account for non-independence of survival and fecundity among age classes a given year. We have clarified this sentence in the revised manuscript (l. 241 p. 12): “Finally, year was added as a fixed (numeric) effect to account for trends. Year was also added as a random (intercept) effect to account for non-independence among age classes, (...).”

In lines 238-242 you use a logit link for fertility, so this implies that you assume that every female will produce at most one calf per year, is this realistic?

>> It has never been observed that Svalbard reindeer can produce more than one calf per year. We thank the reviewer for reminding us to give this important information. In the revised Materials and Methods section, we have clarified this point (l. 249 p. 12): “Likewise, we ran a model of similar structure for fecundity F of each age class i at year t , as females produce at most one calf per year”.

In lines 244-249 you explain that you pooled ages into general age-classes. How do you think this can affect your simulations? I ask because it is well known that reducing ages into age classes dampens down the variance in population growth rates...

>> In the IPM study (Lee et al. 2015), it was found that some ages exhibit similar survival and fecundity rates, and they were therefore pooled into age classes. Based on these findings, we do not believe that the age pooling has a noteworthy effect on our results. Note that, to model the dynamics of the population, we have of course used ages (and not age classes) in the simulations.

In lines 254-258 you describe the covariance matrix for the vital rates, which you base on the random year effects and the residuals. How do you populate this covariance matrix, by calculating the covariation on the random terms between vital rates?

>> Exactly. We clarified this confusing sentence by replacing “built” with “estimated” (l. 270 p. 13).

Lines 265-269: if I understand well and as you write it in eq (4), the variance of the demographic stochasticity is not inversely proportional to the population size, isn't it?

>> *The effect of demographic stochasticity on population growth is indeed inversely proportional to the population size. This is a direct consequence of survival and fecundity being realizations of stochastic processes (in our case, they are both modelled as binomial). In populations with few individuals, the success of each individual has a large effect on the mean success of the population (imagine a population of just two individuals – it's not that unlikely that they will both happen to have a bad year at the same time, with huge consequences for the population). In large populations this will tend to even out and the chance events that affect individuals independently (i.e., demographic stochasticity) will have much less influence on the population as a whole (it is extremely unlikely in a population of e.g. 1500 individuals that they will all happen to have a bad year at the same time). Thus, the demographic variance is not affected by population size directly, but the effect of the demographic variance on the population growth rate is inversely proportional to population size. We have added to l. 280 p. 14 to clarify what demographic stochasticity is in this context: “demographic stochasticity (i.e., chance events that affect individuals independently)”.*

Line 278, when you describe the interval, replace the colon by a coma (i.e. [2, 13]).

>> *The change has been made.*

In lines 287-289 if I interpret correctly these two sentences, you did not simulate the number of calves and 11+ year olds? Why?

>> *We thank the reviewer for this comment. The number of individuals has been estimated for all ages. We have clarified this point by replacing $N_{sim\ 11,t}$ with $N_{sim\ j,t}$ to make the statement more general (see l. 303 p. 15).*

In lines 300-302 you use F in reference to a distribution while you also use it when you refer to fecundity. For clarity, I would suggest that you change this.

>> *We changed F to D.*

In Lines 303-309 you say that you sampled from a Beta distribution. I'm curious, why a Beta distribution. Are the ROS values bound between 0 and 1? Also, if you simply randomly sampled from a Beta distribution the whole explanation of the inverse sampling in lines 300-302 is not necessary.

>> *We are not sure we understand what the reviewer means here. It is the cumulative ROS distribution that is bounded between 0 and 1, that is why we used a Beta distribution and changed the shape of the distribution by adjusting α_1 and α_2 . We clarified this point in the revised manuscript (l. 318 p. 15): “Thus, because the cumulative ROS distribution is bounded between 0 and 1, we used a beta distribution $f(U) \propto U^{\alpha_1-1}(1-U)^{\alpha_2-1}$ where α_1 and α_2 are shape parameters, to simulate five different ROS scenarios.”*

Lines 315-317: This passage is a bit confusing. Do you mean that you started all your simulated populations with the population numbers of 2014? Also, 100 simulated populations seems a bit low...

>> *All the simulations started with the estimated mean age-specific population sizes of 2014. 100 simulations have been performed for each 9,090 population model (i.e. each 9,090 posterior sample), for each ROS scenario (n=5), and for a 100-year long period, resulting in 4,545,000 population time-series (454,500,000 population sizes) in total. We clarified this point in the revised manuscript (l. 329 p. 16): “Using the estimated mean age-specific population sizes in 2014 and each of our 9,090 age-structured population models (i.e. based on 9,090 posterior samples, see Modelling step 2), we simulated for each of the five ROS scenarios (i.e. distributions) 100 stochastic population size trajectories for 100 year projections. This resulted in 4,545,000 population size time-series (454,500,000 population sizes) in total.”*

In Extended data figure 4, you include error bars to the observations. Does that mean that you are accounting for observation errors?

>> *As we performed the analyses directly on the posterior samples, we accounted for uncertainty in the estimated parameters (e.g. 9,090 values of annual age-specific survival and fecundity were used). Importantly, the IPM used to estimate annual age-specific demographic rates and population sizes indeed accounted for observation errors (see Lee et al. 2015). We hope that the new figure 2 summarizing the different methodological steps now clarifies how we accounted for sources of uncertainty around the estimated parameters throughout the analyses.*

Reviewer #2 (Remarks to the Author):

This paper examines the population impact of an extreme environmental stressor (rain on snow, which causes icing conditions making subsequent feeding more difficult) on the dynamics of a reindeer population. Despite substantial increases in overwinter mortality associated with ROS, the authors use models parameterized for this population to project that increasing frequency of ROS (a likely outcome of climate change) will lead to a *reduction* in population fluctuations and extinction risk. This appears to be the result of a combination of demographic and density-dependent buffering, which has been examined in theoretical models, but not (to my knowledge) in particular populations.

The authors performed a thorough and well-thought-through analysis, and I detected no fundamental flaws.

>> *We thank the reviewer for her/his positive evaluation of our paper.*

However, there are a number of areas where the methods and results are not presented in sufficient detail to fully rule out potential challenges to the conclusions. I detail these below. If the analysis stands up to this increased scrutiny, then I think it will be a valuable contribution to the literature on ecological responses to a changing climate.

>> *We provide below a detailed point-by-point response to the reviewer.*

The key relationships that lead to the result described above were not observed directly in the population. Indeed, a key factor in the mechanism--density dependence in the absence of ROS--is not supported by the population data (the curves for high, medium and low density intersect at ROS=0 in fig 1c; the main effect of N on population growth rate is indistinguishable from zero (Table ED1).

>> *The first analysis assessing the effect of population size and rain-on-snow (ROS) on population growth rate is a first preliminary (and explorative) analytical step, which does not assess demographic mechanisms but provides a basis for building the population model based on vital rates. We clarified this point in the revised manuscript (l. 74 p. 4): “As a preliminary analysis, we first explored the impact of ROS and population density on annual population growth rates (...).”; (l. 82 p. 5): “Second, based on these preliminary findings, we explored the underlying demographic mechanisms by modelling annual age-specific survival and fecundity rates, obtained for 9,090 posterior samples from the IPM²⁷, as a function of weather and population size (...).” We also clarified these different methodological steps in a new Figure 2.*

Density-dependence is, according to our demographic models, expected to be weak in the absence of ROS (see Figure 3 in the previous version, i.e. Figure 4 in the revised version). Nevertheless, it is not surprising that some of the estimates from a simple multiple regression of the “observed” population growth rates, as shown in Figure 1 and Table ED1, show some uncertainty and deviations from what is theoretically expected. Yes, we agree that the estimated ROS regression lines for the three density classes do not intersect exactly where they are expected to intersect, and that the parameter estimate for N (i.e. at zero ROS) in Table ED1 is very uncertain, but again, this is not a mechanistic model of demographic rates. We feel that the overall patterns from this simple preliminary model, which was based on 20 data points of population growth rates and only provided a basis for assessing demographic mechanisms, are overall convincing and in line with our other results.

Thus, the analysis depends critically on a previously parameterized model of the population, which is described elsewhere.

>> *We believe this comment is based on a misunderstanding and have included a schematic figure (new Figure 2) to clarify, see also reply further below.*

As a consequence, the ability to use the analysis to draw conclusions about the Svalbard reindeer herd (as opposed to more theoretical conclusions based on a “realistic” population model) requires that the model is an accurate and precise representation of the population processes that are important in the current context. The publications describing the model have been peer reviewed, and I have no reason to question the model (although my experience is that journal peer review provides no guarantee that a published ecological model is reliable); but any model is an approximation to reality, and its utility can only be judged in the context of a particular question. Thus, my question for the authors is how they have evaluated the model’s suitability for the present purpose.

The analysis summarized by ED Fig 4 provides some evidence that the model tracks the year-to-year variation in population abundance (although the confidence intervals for the observed

abundances are generally quite large, resulting in low power, and in some years the model prediction deviates substantially from the point estimate of abundance).

>> *The correlation between observed versus estimated population sizes is convincingly high ($r = 0.89$). It is also noteworthy that beyond population growth rates, our model provides estimates of annual age-specific survival and fecundity rates close the observed values (please, see ED Figures 1 and 2).*

However, it would be helpful to have a qualitative assessment of whether any structural decisions made during the original model development might be more consequential in the current context than the original one, and explicit treatment of the effects of parameter uncertainty (it might be that the posterior samples in the second step are doing this, but it is unclear; see below).

>> *Lee et al. (2015) developed the IPM for our Svalbard reindeer study population (1994-2014). This IPM (updated in Bjørkvoll et al. (2016)) combined several different sources of data (capture-mark-recapture data, population count data and harvesting data) and accounted for observation error, environmental and demographic stochasticity and age structure, thus providing robust estimates of annual age-specific demographic rates (i.e. survival and fecundity) and population sizes. However, this model did not incorporate any environmental or density effects. The role of the IPM in this current study was simply to provide estimates of annual demographic rates and population sizes as “data” (please, see the new Figure 2 for some clarification). We have good reason to believe that these estimates are accurate, based on comparison of estimates from the original model with data from a more rigorous population count of the same population in those same years (see Lee et al. 2015 for more details). These independent count data has since been incorporated into the IPM (in Bjørkvoll et al. (2016)), further improving the estimates. In the current study, we have used the 9,090 posterior samples from the IPM (meaning 9,090 estimates of annual age-specific demographic rates and population sizes), to assess how weather and population size affect demographic rates and, thereby, population growth. We agree that the previous version of the manuscript was not clear enough in explaining how the different modelling parts fit together and what role the previously published IPM played. To make it clearer, we have added a new figure (Figure 2) that summarizes the different methodological steps and shows that the IPM is only a preliminary model allowing us to get accurate and precise estimates of demographic rates for the reindeer population.*

From a cursory examination of the cited papers, it appears that ROS is not in the original IPM. Please explain how ROS data was matched with the IPM outputs to allow the regressions.

>> *As explained above, there was no environmental covariates in the IPM. The IPM simply allows us to get accurate and precise estimates of annual demographic rates. In the present study, based on those annual estimates provided by the IPM, we tested an effect of annual ROS on annual age-specific survival and fecundity rates (please, see Eqn. (1) and (2)). We hope that the new Figure 2 will clarify the fact that no environmental data was included in the IPM itself, and that we tested an effect of ROS on each posterior sample (i.e. on 9,090 annual age-specific demographic rates). This generated 9,090 ‘population models’ of weather and density effects.*

What constitutes a posterior sample from the IPM? A single year of vital rate estimates and associated covariates? A full time series of such values?

>> *From the IPM, we got 9,090 posterior samples. One posterior sample consists of annual age-specific demographic rates (i.e. survival and fecundity) and population sizes from 1994 to 2014. For each of these 9,090 posterior samples, we estimated an effect of ROS and population size from Eqn. (1) for survival estimates and from Eqn (2) for fecundity estimates. We have made this point clearer in the manuscript (l. 223 p. 11): “We obtained 9,090 posterior samples from the IPM^{27,30}, each of them consisting of annual age-class specific demographic rates (i.e. survival and fecundity) and population sizes from 1994 to 2014.”*

How were models in equations 1 and 2 estimated? If ROS came from observed data, then there really are only 20 independent observations, and there is a risk of pseudoreplication if the posterior samples are treated as independent. I would imagine that the ideal approach would be for each posterior sample to be a full time series, and the regression equations were estimated separately for each sample. The posterior distributions on the parameters of eqs. (1) and (2) would then represent a combination of uncertainty due to finite time series and uncertainty stemming from both stochasticity and parameter uncertainty in the IPM. Is this what was done? If so, please describe it explicitly. If not, please describe and justify the approach.

>> *We thank the reviewer for her/his comment. We have effectively fitted the regression equations for each posterior sample separately. Following the reviewer's suggestion, we wrote in the revised manuscript (l. 225 p. 11): “For each of these posterior samples, we investigated the effects of weather and population density on survival and fecundity, generating in total 9,090 ‘population models’. First, the effects of weather and population density on survival S of each age class i at year t were estimated separately for each posterior sample (Fig. 2, Extended Data Fig. 1).”; (l. 247 p. 12): “The posterior distributions of the parameters of Eqn. (1) shown in Extended Data Table 2 represent a combination of uncertainty due to finite time series, uncertainty stemming from stochasticity, and uncertainty in the IPM.”; (l. 255 p. 13): “Again, the posterior distributions of the parameters of Eqn. (2) shown in Extended Data Table 2 represent a combination of uncertainties (see above for survival).”*

I don't understand the argument justifying the form of ROS' (ll. 225-7). The direction of the effect depends on the sign of the regression coefficient, which is unconstrained. Given that this introduces a very specific form of nonlinearity that contrasts with linear direct effect of N , this needs conceptual or empirical justification.

>> *We thank the reviewer for her/his comment. We agree that this statement “which ensures that the effect of increasing ROS (or increasing density) on survival will never turn positive” was too strong and that the direction of effects may depend on the sign of the regression coefficient. We have now changed the statement and we have also added some explanation why we choose this specific form:*

(l. 231 p.12): “We fitted a density-dependent ROS effect²⁰ (cf. Fig. 1 c) using the form $ROS_t' = ROS_t \times e^{k \times N_{posthunt,t}}$ (see ²¹ for a similar approach). This form of the interaction effect ensures that the effect of ROS is strictly negative or positive (depending on the fitted coefficient of ROS') for all values of $N_{posthunt}$. In contrast, the simpler and more common specification of an interaction effect, such as $ROS_t' = k \times ROS_t \times N_{posthunt,t}$,

would implicitly lead to a switch in sign of the effect of ROS at some level of N, and the stronger the interaction effect, the more likely it is that the change of sign is well within data range.”

>> *Because of the logit-link and the back-transformation using the inverse-logit transformation to the probability scale, which is bounded between 0 and 1, the model is inherently nonlinear. We cannot think of any a priori ecological justification for fitting a strictly linear effect of population density on logit(probability of survival) and logit(probability of having a calf). The sign of the effect of ROS is not expected to change with changing density. Our way of specifying a density-dependent ROS effect gave priority to this, making sure that the model did not give ecologically unreasonable estimates for the effect of ROS.*

Furthermore, the posterior CI for k spans zero, indicating that the interaction between N and ROS could be either positive or negative.

>> *The referee points out that posterior CI for parameter k spans zero. The estimates provided in Extended Data Table 2 correspond to the posterior distributions of the parameters of Eqn. (1) and (2). Posterior correlations of parameters are not shown and some of the parameters have a sizeable posterior correlation. In particular, the estimates of effects of N and estimates of k are correlated in the posterior distribution ($p = 0.6$ and $p=0.75$ for survival and fecundity respectively). The joint uncertainty of parameters (correlation among parameters) is accounted for in the analyses and simulations by using sets of parameters drawn from the posterior distribution, and our results thus reflect this uncertainty. We have now added figures in the appendix (ED Figs. 8 and 9) showing how predicted fecundity and survival varies with increasing population density for extreme values of ROS across different sets of parameters from the posterior distribution. These figures illustrate that main patterns, and effects are rather consistent across different sets of parameters in the posterior distribution. Even negative values of k tend to give reasonable predictions, and this is probably due to strong dependency among estimated parameters. Hence, from a biological viewpoint, we view these results as fully relevant.*

It doesn't appear that the simulations leading to figs 2 and 3, and to the primary result of the paper, account for uncertainty in the parameter estimates shown in Table ED2. Especially given that some of the estimates (notably the interaction between ROS and N) have credible intervals that span zero, I think that it would be critical to evaluate how robust the results are to that uncertainty.

>> *The estimates provided in ED Table 2 correspond to the posterior distributions of the parameters of Eqn. (1) and (2). But as explained in one of our previous responses to the reviewer, we have fitted the regression equations Eqn (1) and (2) for each posterior sample separately, so ED Table 2 simply summarizes the results for the 9,090 posterior samples. Uncertainty has thus been accounted for in the analyses and we hope the response to previous comments will clarify this point. Figure 2 (now Figure 3) shows simulations from randomly picked posterior samples. Figure 3 (now Figure 4) shows 10,000 randomly sampled population sizes/growth rates across the simulations from all posterior samples.*

>> *We are aware that some of the estimates have credible intervals that span zero. However, this is not the case for the N effect, nor the N-dependent ROS effect, i.e. both are “significant” for certain age classes. Also, the fact that predictors affect survival and fecundity rather similarly (and in the direction as expected) provides further confirmation that we have a decent model for climate-density effects in our population.*

Is the residual variance (attributed to other sources of environmental stochasticity) kept constant with increasing ROS frequency? Or is it rescaled to keep total variability constant? If the latter, that could contribute to result of lower population variability/extinction risk at high ROS frequency.

>> *ROS distribution has been changed but the variance-covariance remained constant.*

The variable "ROS" is defined and used inconsistently. In figure 1, ROS is shown as a quantity measured in mm. However, in the data description (ll. 195-6), ROS is defined as the log of (amount in mm +1). It is unclear which is being used to define ROS' at l. 226. Furthermore, in the simulation analysis, the authors refer to "ROS frequency scenarios" (e.g. 11. 103-111). As best I can tell from fig ED 6, these scenarios have different mean ROS intensity, along with shifts in the tails of the ROS intensity distribution. These inconsistencies and impressions in notation make the work difficult to follow.

>> *We have carefully checked all the terms (“ROS” and similar) and made sure to be consistent throughout the revised manuscript. In particular, we have changed the labels of the axes in Figure 1, Figure 3 (old Figure 2), Figure 5 (old Figure 4), Figure 6 (old Figure 5). We changed “ROS frequency scenarios” to “ROS scenarios” throughout the text.*

The parameter values in table ED 2 are very difficult to interpret, largely because the “intercept” represents the demography of the first age class extrapolated back to the year 0 (nearly 2000 years before the beginning of the time series. Because of the positive coefficient for year, this results in rates on the probability scale of essentially zero, and greatly inflates the apparent uncertainty in first age class demography. I recommend either centering the year variable or rescaling it so that “year 0” is the beginning of the time series. It would also be helpful to have a version of the table that reported the age-class estimates directly (both the base survival/fecundity and the impacts of ROS’), rather than as differences from the first age class. This would reveal, for example, that the estimated effect of ROS’ on age class 3 survival has a CI that spans zero.

>> *We agree with the reviewer that table ED 2 was difficult to interpret. In the revised manuscript, as suggested by the reviewer, the year effect has been centered and the estimates have been reported for all age classes.*

The deterministic model results provide valuable insights, thank you for including this analysis. However, it seems to me that the result is not just (or even primarily) the fact that prime age individuals are less sensitive to ROS (which is only true for survival, incidentally; the fecundity ROS coefficients are more negative for prime age classes), but also due to the relaxation of density dependence. This should be discussed more thoroughly.

>> *Thanks to the reviewer for addressing this. We agree, but we also feel that the importance of density dependence is already discussed quite well in the manuscript. However, to further stress this, we added or changed: (l. 122 p. 6): “With no subsequent ROS winter (Fig. 6 a, leftmost panel), the low density (i.e. relaxation of density dependence) and new age structure allow the population to recover towards an asymptotic population size (...);” (l. 116 p. 6): “Provided a rather high initial density characterized by a low proportion of prime-aged animals (...).”; (l. 127 p. 7): “This occurs because the previous year’s crash generated a new population state, with low density as well as large proportion of prime-aged animals, showing little sensitivity to ROS (...).” ; (l. 129 p. 7): “If the second ROS winter is delayed until $t = 2$, the previous year’s recovery in population size induces a slight population decline due to the harsh feeding conditions.” ; (l. 133 p. 7): “Accordingly, if there are several extreme ROS winters in a row, population size converges towards a reduced density of 1000 - 1500 individuals due to relaxation of density dependence (...).”; (l. 144 p. 7): “When ROS is still that rare, ungulate populations are more likely to be in the kind of state (in terms of density, as well as demographic structure) that may lead to a crash when an extreme event occurs.”*

Furthermore, since ROS disproportionately impacts age 1 survival, repeated high ROS years will substantially reduce recruitment to the prime age population. If ROS events are simulated not just twice in a row, but many times in a row, does the population converge to a low-density equilibrium, or does it continue to decline? Also, are the deterministic model results robust to the uncertainty in the parameter estimates?

>> *Yes, we find that the results from the deterministic simulations are robust to the uncertainty (see Figure 6 [old Figure 5] and new Figure 7). Following the reviewer's suggestion, we have included an additional analysis showing population size over time under a scenario with many consecutive ROS winters (new Figure 7). We have modified the manuscript accordingly: (l. 353 p. 17): “We also included a ROS sequence with no second extreme ROS winter, and a sequence with many consecutive extreme ROS winters.”; (l. 133 p. 7): “Accordingly, if there are several extreme ROS winters in a row, population size converges towards a reduced density of 1000 - 1500 individuals due to relaxation of density dependence (Fig. 7).”*

>> *We thank the reviewer for this comment, it was an important analysis to add in the manuscript.*

>> *Please note that we removed ED Fig. 8 from the old version of the manuscript, as it was not discussed, and, in hindsight, not considered important for the paper.*

Reviewers' Comments:

Reviewer #1:

Remarks to the Author:

I believe that the authors have addressed all my comments. I still think that this is a very valuable study.

Fernando Colchero

Reviewer #2:

Remarks to the Author:

In my prior review, my concerns were around lack of clarity in the presentation of the models and methods. Thank you for the careful attention to my previous comments; I have no further concerns with regard to those issues.

I do, however, find myself somewhat confused by the meaning of "fecundity" ($F_{I,t}$) in eqs. 2 and 4. From Eq. 4, it appears that the fraction of individuals of, say, age 3 giving birth in a given year is determined by characteristics of 2-year-olds a year previously. Is F actually an estimate of November pregnancy rate? How is it possible that pregnancy in autumn is affected by environmental conditions the following winter (which is what I understand ROS_t to be)? If $F_{2,t}$ is just the fraction of 3-year-olds with calves in year $t+1$, why not index it as $F_{3,t+1}$? Please provide a precise definition for this term and, if appropriate, adjust the indexing.

REVIEWERS' COMMENTS:

Reviewer #1 (Remarks to the Author):

I believe that the authors have addressed all my comments. I still think that this is a very valuable study.

Fernando Colchero

Thank you.

Reviewer #2 (Remarks to the Author):

In my prior review, my concerns were around lack of clarity in the presentation of the models and methods. Thank you for the careful attention to my previous comments; I have no further concerns with regard to those issues.

Thank you.

I do, however, find myself somewhat confused by the meaning of “fecundity” ($F_{\{1,t\}}$) in eqs. 2 and 4. From Eq. 4, it appears that the fraction of individuals of, say, age 3 giving birth in a given year is determined by characteristics of 2-year-olds a year previously. Is F actually an estimate of November pregnancy rate?

How is it possible that pregnancy in autumn is affected by environmental conditions the following winter (which is what I understand ROS_t to be)? If $F_{\{2,t\}}$ is just the fraction of 3-year-olds with calves in year $t+1$, why not index it as $F_{\{3,t+1\}}$? Please provide a precise definition for this term and, if appropriate, adjust the indexing.

Thanks for pointing this out. In some ways we agree that our $F_{(2,t)}$ could logically have been called $F_{(3,t+1)}$ instead, as it does represent the probability of a 2-year-old at time t having a calf at heel at time $t+1$. However, because survival of a 2-year-old from t to $t+1$ is indexed as $S_{(2,t)}$, and these two probabilities are influenced by climate and population size at the same time step (t) we believe our current indexing scheme is the least likely to cause confusion. We have added a sentence following Equation (4) to better explain this term.

**Best regards,
Brage B. Hansen & Marlene Gamelon**